# Mesothelioma and Radical Surgery 2 (MARS 2): protocol for a multicentre randomised trial comparing (extended) pleurectomy decortication versus no (extended) pleurectomy decortication for patients with malignant pleural mesothelioma

Eric Lim,[1] Liz Darlison,[2,3] John Edwards,[4] Daisy Elliott,[5] D A Fennell,[6] Sanjay Popat,[7] Robert C Rintoul,[8] David Waller,[9] Clinton Ali,[10] Andrea Bille,[11] Liz Fuller,[12] Andreea Ionescu,[13] Manjusha Keni,[14] Alan Kirk,[15] Pek Koh,[16] Kelvin Lau,[9] Talal Mansy,[17] Nick A Maskell,[18,19] Richard Milton,[20] Dakshinamoorthy Muthukumar,[21] Tony Pope,[22] Amy Roy,[23] Riyaz Shah,[24] Jonathan Shamash,[25] Zacharias Tasigiannopoulos,[26] Paul Taylor,[27] Sarah Treece,[28] Kate Ashton,[29] Rosie Harris,[29] Katherine Joyce,[29] Barbara Warnes  ,[29] Nicola Mills,[5] Elizabeth A Stokes  ,[30] Chris Rogers,[29] On behalf of MARS 2 Trialists

**Correspondence to**
Eric Lim; e.lim@rbht.nhs.uk

## ABSTRACT

**Introduction** Mesothelioma remains a lethal cancer. To date, systemic therapy with pemetrexed and a platinum drug remains the only licensed standard of care. As the median survival for patients with mesothelioma is 12.1 months, surgery is an important consideration to improve survival and/or quality of life. Currently, only two surgical trials have been performed which found that neither extensive (extra-pleural pneumonectomy) or limited (partial pleurectomy) surgery improved survival (although there was some evidence of improved quality of life). Therefore, clinicians are now looking to evaluate pleurectomy decortication, the only radical treatment option left.

**Methods and analysis** The MARS 2 study is a UK multicentre open parallel group randomised controlled trial comparing the effectiveness and cost-effectiveness of surgery—(extended) pleurectomy decortication—versus no surgery for the treatment of pleural mesothelioma. The study will test the hypothesis that surgery and chemotherapy is superior to chemotherapy alone with respect to overall survival. Secondary outcomes include health-related quality of life, progression-free survival, measures of safety (adverse events) and resource use to 2 years. The QuinteT Recruitment Intervention is integrated into the trial to optimise recruitment.

**Ethics and dissemination** Research ethics approval was granted by London – Camberwell St. Giles Research Ethics Committee (reference 13/LO/1481)

### Strengths and limitations of this study

► (Extended) pleurectomy decortication is currently offered to patients with mesothelioma on the United Kingdom National Health Service, but it is unknown whether it is a clinically beneficial or cost-effective treatment option. MARS 2 is the first randomised controlled trial to compare this type of surgery with no surgery in this patient population.

► Surgical quality assurance measures will be implemented to ensure that the intervention will be delivered at centres with expertise.

► Patients may come with a pre-conceived perception that surgery will be beneficial, which can lead to crossovers (ie, patients allocated to no surgery may go on to seek surgery elsewhere). The integrated QuinteT Recruitment Intervention supports recruitment staff in responding to patient preferences and conveying balanced information.

► It is not possible to blind participants or the study team, but the primary outcome (survival) is objective.

► Patient pathways vary at different sites. Some flexibility has been worked into the protocol to allow for this.

on 7 November 2013. We will submit the results for publication in a peer-reviewed journal.

**Trial registration numbers** ISRCTN—ISRCTN44351742 and ClinicalTrials.gov—NCT02040272.

## INTRODUCTION

In the UK, approximately 2500 patients are diagnosed each year with pleural mesothelioma,[1] a treatment-resistant and lethal cancer of the membranes lining the outer surface of the lung and the inside of the chest wall primarily due to asbestos exposure. Deaths are increasing yearly and are estimated to peak this year.[2] So far, most treatments have proven ineffective. The current standard of care, consisting of 4 to 6 cycles of platinum and pemetrexed chemotherapy, as recommended by the National Institute for Health and Care Excellence (NICE),[3] has been associated with only an additional 3 months of survival.[4] As the median survival for patients with mesothelioma is 12.1 months,[4] surgery to remove as much of the disease as possible remains an important consideration to improve survival and/or health-related quality of life (HRQoL).[5]

Pleurectomy decortication is the most common surgical procedure for mesothelioma worldwide and is defined as parietal and visceral pleurectomy to remove all gross tumour without diaphragm or pericardial resection.[6] Extended pleurectomy decortication can also be carried out, when parietal and visceral pleurectomy is undertaken to remove all gross tumour, including the resection of the diaphragm and/or pericardium. In the document we use the term (extended) pleurectomy decortication to refer to either of the two procedures. The other main types of surgery for mesothelioma are extra-pleural pneumonectomy, which is defined as en bloc resection of the parietal and visceral pleura with the ipsilateral lung, pericardium and diaphragm (in cases where the pericardium and/or diaphragm are not involved by tumour, these structures may be left intact); and partial pleurectomy, which is the partial removal of parietal and/or visceral pleura for diagnostic or palliative purposes but leaving gross tumour behind.[6]

So far, no advantage, in terms of survival, has been observed with any type of surgery in randomised controlled trials (RCTs) conducted to date. The MARS feasibility study (ISRCTN95583524), a trial of extra-pleural pneumonectomy with adjuvant haemothorax irradiation, concluded that surgery was unlikely to offer either an improvement to survival or HRQoL and possibly harmed patients.[7] MesoVATS (ISRCTN34321019) concluded that partial pleurectomy did not improve survival, although it showed that patients in the better prognostic group had improved HRQoL after 6 months.[8]

Suitable patients, both in the UK and internationally, are currently offered pleurectomy decortication as it is considered to carry less morbidity compared with the more extensive extra-pleural pneumonectomy but still achieves complete macroscopic resection which partial pleurectomy does not.[9–11] However, we do not know if (extended) pleurectomy decortication in conjunction with chemotherapy will improve survival compared with the current standard of care (chemotherapy alone). In the absence of RCTs, (extended) pleurectomy decortication may continue to be offered despite a lack of high-quality evidence of clinical efficacy or any evidence on cost-effectiveness.

### Aims and objectives

MARS 2 is a UK-wide multicentre RCT which will test the hypothesis that (extended) pleurectomy decortication and chemotherapy is superior to chemotherapy alone with respect to overall survival for patients with pleural mesothelioma.

Specific objectives are to estimate:
A. The difference between groups in overall survival.
B. The difference between groups with respect to a range of secondary outcomes including HRQoL, progression-free survival and measures of safety (adverse health events).
C. The cost-effectiveness of (extended) pleurectomy decortication compared with no surgery.

## METHODS AND ANALYSIS

### Trial design

MARS 2 is a multicentre, non-blinded parallel two-group, pragmatic RCT of surgery and chemotherapy versus chemotherapy alone for suitable patients with mesothelioma.

An internal pilot funded by Cancer Research UK (award ref: C27967/A15895) and coordinated by the Papworth Trials Unit Collaboration demonstrated the feasibility of recruitment across 14 medical sites and 2 joint medical and surgical sites of excellence, as the target of 50 participants recruited within a 24-month period was achieved.

Since the end of the pilot phase in December 2016, an additional eight medical, one surgical, and two joint medical and surgical sites have been opened for the full trial. In addition, the full trial will provide recruiting sites with the support of an integrated QuinteT Recruitment Intervention (QRI)[12–14] to optimise recruitment and retention.

### Setting, centre and surgeon eligibility

This study is taking place in National Health Service (NHS) secondary care centres, including teaching and district general hospitals.

To be eligible as a medical site, the centre must:
1. Be an NHS Trust with access to a multidisciplinary team (MDT) to discuss patients with mesothelioma.
2. Have a track record of treating patients with mesothelioma.

To be eligible as a surgical site, the centre must:
1. Be an NHS Trust with an established mesothelioma MDT.
2. Have a minimum of two named mesothelioma surgeons participating in the trial.

All surgeons participating in the full trial must be accredited by (1) self-reporting a minimum of five cases in which they have performed (extended) pleurectomy decortication, (2) observing the procedure being undertaken at an established MARS 2 surgical site, (3) having a surgeon from the pilot phase observe their first MARS

2 procedure undertaken and (4) having one randomly selected MARS 2 operation between procedures 5 and 10 observed by a surgeon from the pilot phase to ensure fidelity.

Patients from all medical (only) sites are referred to a trial-accredited surgical site for CT assessment of eligibility, further discussion about the study and surgery (if randomised to this group).

## Trial population

The target population are patients with a diagnosis of epithelioid, sarcomatoid or biphasic mesothelioma. Patients will be eligible to take part if ALL of the following apply:
► Adult aged ≥16 years of age.
► Tissue (cytology or histology) confirmed epithelioid, sarcomatoid or biphasic mesothelioma, as reviewed by MDT to be of sufficient certainty to recommend chemotherapy as treatment.
► Disease confined to one hemithorax based on CT assessment.
► Disease deemed surgically resectable by a surgeon at a MARS 2 surgical site.
► Deemed fit for surgery by a surgeon at a MARS 2 surgical site.
► Capacity to provide written informed consent to participate in the trial.

Patients will not be eligible if they have:
► Severe shortness of breath (Eastern Cooperative Oncology Group status ≥2, or preoperative FEV1 or TLco less than 20%).
► Severe heart failure (NYHA III or IV, or ejection fraction less than 30% by echocardiogram).
► End-stage kidney failure requiring dialysis.
► Liver failure (eg, encephalopathy and/or coagulation abnormalities).
► Any other serious concomitant disorder that would compromise participant safety during surgery.
► Prisoner.
► Patient lacks capacity to consent.
► Existing co-enrolment in another interventional study that aims to improve survival.

## Patient approach, consent and randomisation

The local research team at the medical site will take written informed consent from participants. In addition to the main study, the team may also seek consent for audio-recording of consultations and participation in interviews, for QRI purposes. Participants will then receive two cycles of chemotherapy (standard care) and have a further CT scan to confirm eligibility (ie, disease still resectable) before being randomised, using a secure web-based randomisation system (Sealed Envelope https://sealedenvelope.com).

Participants will be randomised in a 1:1 ratio. Minimisation (with a random component) will be applied for selected baseline variables (age, performance status and cell type) that influence survival, in addition to

stratification by recruiting site to ensure that the cohorts are as balanced as possible.

## Trial interventions

Patients will be randomised to receive one of the following interventions:
► *(Extended) pleurectomy decortication and chemotherapy*: two cycles of platinum and pemetrexed chemotherapy followed by surgery and then up to four cycles of the same chemotherapy.
► *Chemotherapy alone (control intervention)*: up to six cycles of platinum and pemetrexed chemotherapy alone (current standard of care).

The trial schema is illustrated in figure 1.

After randomisation, any changes in the choice of chemotherapy, addition of other agents or entry into therapeutic trials (eg, immunotherapies) will be permitted for patients with progressive disease. At the time of trial design, there was no national consensus on postoperative prophylactic radiotherapy, so it was decided that irradiation to thoracic procedure sites may be undertaken for MARS 2 patients. Patients in both groups can also receive further surgery, including thoracic, if it is without radical intent. The aim is to conduct a pragmatic trial while closely monitoring uptake of additional therapies, studies or surgeries in order to account for them in the trial analyses, if required.

## Primary and secondary outcomes

The primary outcome is survival, calculated from randomisation date (randomisation occurs after the first two cycles of chemotherapy). All participants will be followed up to the end of the trial (minimum of 2 years after randomisation).

Secondary outcomes have been selected to assess the efficacy of the two approaches. Secondary outcomes are (1) progression-free survival to the end of the trial (minimum of 2 years after randomisation); (2) serious adverse health events to 2 years after randomisation; (3) disease-specific and generic HRQoL using the following validated questionnaires—European Organisation for Research and Treatment of Cancer Quality of Life Questionnaire (EORTC QLQ-C30), to assess the HRQoL of patients with cancer, and EuroQol EQ-5D-5L,[15 16] a widely used generic measure of HRQoL (both of these will be measured at baseline, pre-randomisation, and 6 weeks, 6, 12, 18 and 24 months post-randomisation); and (4) healthcare resource use to the end of the study: chemotherapy cycles and initial surgical admission (for chemotherapy plus surgery group), and further resources measured at 6 weeks post-randomisation then every 6 months, with a final follow-up at the end of the study if not followed up in the previous 4 months.

## Data collection

The schedule for data collection for the study is shown in table 1. Data will be collected onto purpose-designed case report forms (CRFs) and participant-completed

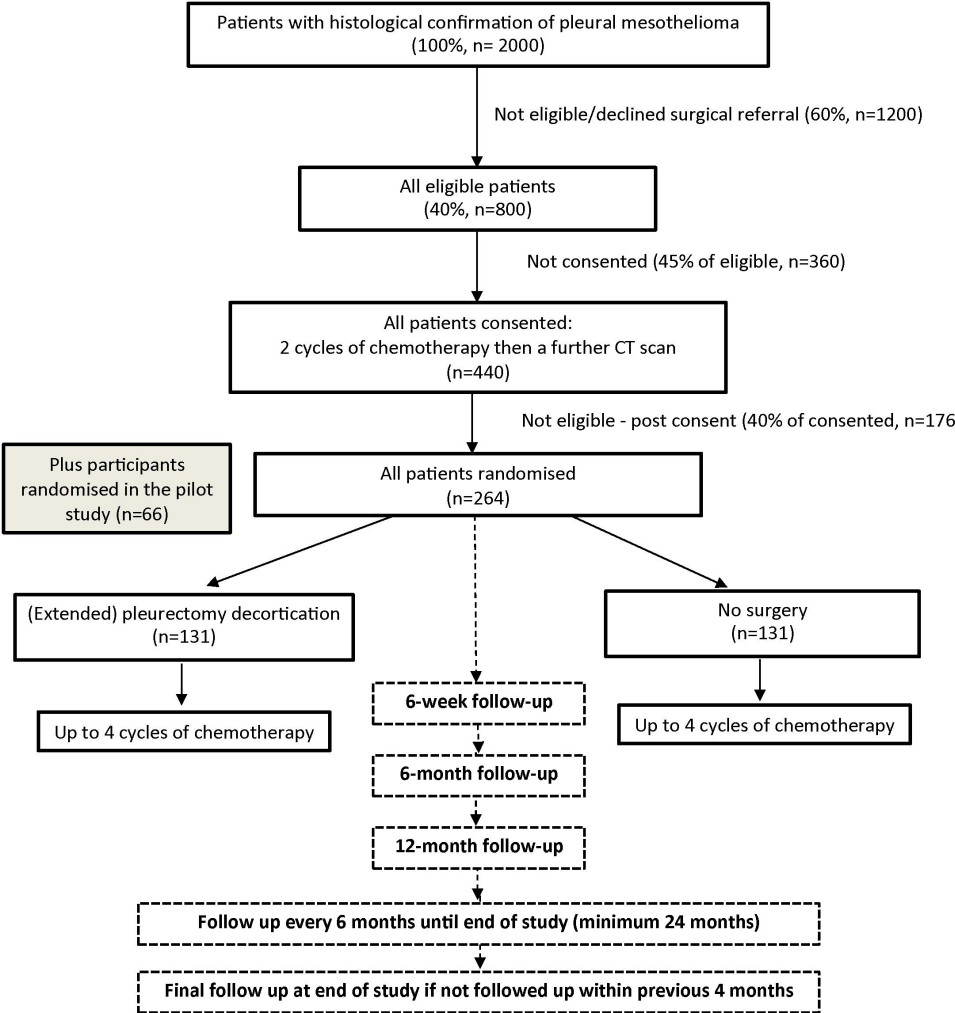

**Figure 1** Trial schema showing the recruitment pathway for the MARS 2 study.

questionnaires and entered onto a bespoke database for data cleaning and analysis. Access to the database will be via a secure password-protected web interface hosted on an NHS server. Data about adverse events will be collected and reported in accordance with sponsor's and regulatory requirements.

### Risk of bias
Participants and clinical personnel cannot be blinded to allocation due to the nature of the study intervention. However, standard local protocols will be followed in terms of patient care. The patient information leaflet and conversations with MARS 2 site staff will describe and balance the potential benefits and risks of both having and not having surgery. Therefore, this approach will reduce participant's expectations that one or other treatment protocol will lead to a more favourable result.

In addition, the study's primary outcome is an objective measure (survival), and clear definitions of each secondary outcome measure will be provided to trial personnel. The HRQoL follow-up questionnaires may be more at risk of bias than other measures, but patients will not have had this surgery previously and as such should not have any expectation regarding its effect on

their HRQoL. Missing outcome data will be minimised, as survival and progression-free survival data can be obtained from hospital records. Losses to follow-up will be minimised by maintaining regular contact with participants (by telephone and post) to complete follow-up questionnaires. Non-adherence to randomised allocation will be documented. Bias in the reported results will be minimised by having pre-specified outcomes in the trial protocol and a pre-specified analysis plan.

### Sample size
The total sample size has been set at 328 participants (164 per group). The patients randomised in the pilot trial will contribute to the total sample size. The study will have 80% power to detect a HR of 0.7 at 5% statistical significance (two-sided), modelled on a published assumption of a median survival time of 16.8 months in patients with mesothelioma who were fit enough to receive surgery, but did not have it[17] and allowing for 10% cross-over from the medical to surgery groups (as noted in previous trials such as MARS).[7] Cross-over will be minimised through instruction (ie, recruit only patients who have equipoise from the outset) and education.

**Table 1** Data collection

| | Pre-randomisation | | | | Post-randomisation | | Follow-up | | | | | |
|---|---|---|---|---|---|---|---|---|---|---|---|---|
| | Screening | Baseline | 2 cycles of chemotherapy | End of chemotherapy cycle 2 | Surgical admission* | Up to 4 cycles of chemotherapy | 6 weeks | 6 months | 12 months | 18 months | 24 months | Every 6 months until end of trial† |
| Screening log | X | | | X | | | | | | | | |
| CT scan | X‡ | | | X | | | | | | | | |
| Informed consent | | X | | | | | | | | | | |
| Demography, medical history, blood test results | | X | | | | | | | | | | |
| Lung function tests | X | X§ | | X | | | | | | | | |
| HRQoL | | X | | X | | | X | X | X | X | X | |
| Chemotherapy treatment given‡ | | | X | | | X | | | | | | |
| Surgery and in hospital postoperative data¶ | | | | | X | | | | | | | |
| Adverse events | | | | | X | | X | X | X | X | X | X |
| Patient-reported resource and health service use | | | | | | | | X | X | X | X | X |

*Patients allocated to surgery and/or receiving surgery only.
†If not within previous 4 months.
‡Previous CT scan to be used (not to be done again specifically for the trial protocol).
§Only one assessment of lung function is needed, so if this has been done prior to screening, there is no need for another test at baseline.
¶Including resource and health service use.

The relative difference of 30% (HR 0.7) was regarded as the minimally important difference for patients and clinicians to choose surgery given the risks of the procedure. The figure was chosen by the trial's patient and public involvement (PPI) group. The possibility that survival could be worse with surgery was also discussed, and a relative difference of 30% was also regarded as an appropriate difference to indicate harm, therefore a two-tailed test for superiority was agreed.

## Patient and public involvement

Patient and public representatives were involved from inception and advised on the trial design of MARS 2, the identification of the choice of the primary outcome and defined the minimally important difference in relative survival.

The study team have continuing engagement with the Royal Brompton Hospital Cancer Consortia PPI group, which consists of patients and carers who have undergone surgery for lung cancer and mesothelioma, to advise on patient-orientated questions that arise from the trial conduct. One patient from the PPI group, a mesothelioma survivor, has agreed to sit on the Trial Steering Committee. The PPI group will also be involved in the dissemination of study results.

## Integrated QRI

Recruitment to RCTs can be challenging,[18] particularly for surgical trials.[19] An integrated QRI will therefore be employed during the main study phase to optimise recruitment and retention. The aim of the QRI is to understand the recruitment process and how it operates in clinical centres, so that sources of recruitment difficulties can be identified, and suggestions made to change aspects of design, conduct, organisation or training.

A multi-faceted, flexible approach will be used to investigate site-specific or wider recruitment obstacles. These will comprise the following:

► Mapping of eligibility and recruitment pathways to collate basic data about the levels of eligibility and recruitment, and identify points at which patients opt in or out of the trial.

► In-depth, semi-structured interviews with a purposive sample of staff members involved with aspects of trial design/management and recruitment across centres, and patients eligible for recruitment to the trial. Interviews will explore participants' perspectives of the trial, views on the presentation of study information, understanding of trial processes (eg, randomisation), and reasons underlying decisions to accept or decline the trial. In addition, interviews with staff and other individuals involved in the trial will explore perspectives on the trial design and protocol, views about the evidence on which the trial is based, perceptions of uncertainty/equipoise for themselves and their colleagues, methods for identifying eligible patients, views on eligibility, and examples of actual recruitment successes and difficulties. Interview topic guides will be used to ensure similar topic areas are covered across interviews, while still providing the scope for participants to raise issues of pertinence to them.

► Audio-recording of consultations between healthcare staff and potentially eligible patients across centres to understand the recruitment process at each centre and to identify and investigate the challenges to recruitment. The QRI researcher will listen to and qualitatively analyse the appointments, documenting instances such as unclear, insufficient or imbalanced information provision and unintentional transferring of clinician treatment preferences to patients.

► Observation of Trial Management Group (TMG) and investigator meetings to gain an overview of trial conduct and overarching challenges (logistical issues, etc).

An account of the anonymised findings from all the data will be fed back to the Chief Investigator and TMG. The data will be used by the QRI team to provide supportive and confidential individual and group feedback to recruiters to help them to communicate equipoise, balance treatment options and explain to patients the benefits and purposes of trial participation, while optimising informed consent.

## Statistical analyses

The data will be analysed for randomised patients according to intention to treat and follow Consolidated Standards of Reporting Trials (CONSORT) guidelines. Analyses will be adjusted for site and for design factors included in the cohort minimisation (eg, age, performance status and cell type).

Survival time and progression-free survival time from randomisation will be compared using survival methods, allowing for censoring of any participant who is either alive or lost to follow-up at the end of the follow-up period. Patient-reported outcome scores (HRQoL EQ-5D-5L and QLQ-C30) will be compared using a mixed regression model and adjusted for baseline measures where appropriate. Changes in treatment effect with time will be assessed by adding a treatment × time interaction to the model and comparing models using a likelihood ratio test. Deaths will be accounted for by modelling survival and HRQoL jointly. Model fit will be assessed using standard methods and alternative models and/or transformations will be explored if appropriate. Treatment differences and 95% CIs will be reported.

Missing data on patient questionnaires will be dealt with according to the scoring manuals. Multiple imputation methods will be used if greater than 5% of cases have missing data, otherwise complete case analysis will be undertaken. Compliance rates will be reported, including the number of participants who have withdrawn from the study, have been lost to follow-up or died. Causes of death for trial participants will be recorded.

Frequencies of adverse events will be described. The proportion of participants experiencing one or more

serious adverse events in the follow-up period will be compared using a generalised linear model.

Two subgroup analyses are planned: (1) comparing primary and secondary outcomes by the experience level of the surgical site; (2) comparing the primary outcome by type of mesothelioma (epithelioid, sarcomatoid or biphasic). An exploratory analysis investigating the effect of surgeon (surgical group only) will be performed for the primary outcome.

No interim analyses are planned. The primary analysis will take place when follow-up is complete for all recruited participants.

## Economic evaluation

The economic evaluation will compare the costs and effects of surgery versus no surgery, following established guidelines as set out by NICE.[20] The within-trial cost-effectiveness analysis will be undertaken from an NHS and personal social services perspective, with a time horizon from time of consent to 24 months post-randomisation. The primary outcome measure for the economic evaluation will be quality-adjusted life years (QALYs), estimated using the EuroQol EQ-5D-5L at each follow-up timepoint.[15 16] Resource use data collection will be integrated into the trial CRFs for chemotherapy cycles and surgery (if applicable, this will include details of the surgical procedure, length of stay in hospital by level of care, and postoperative complications) and be collected at each follow-up timepoint.

Unit costs will be sought to value resource use data, and the total costs per participant calculated. Responses to the EQ-5D-5L will be assigned valuations according to NICE guidance at the time of analysis,[21] and combined with survival to calculate QALYs gained per participant. Missing resource use and EQ-5D-5L data will be handled using multiple imputation methods.[22] From the average costs and QALYs gained in each trial group, the incremental cost-effectiveness ratio will be derived, producing an incremental cost per QALY gained of surgery compared with no surgery. Sensitivity analyses will assess the impact of varying key parameters on baseline cost-effectiveness results. Results will be expressed in terms of a cost-effectiveness acceptability curve, which indicates the likelihood that surgery is cost-effective for different levels of willingness to pay for health gain.

## Ethics and dissemination

The study intervention is already routinely used in the NHS. The pilot study was managed by Papworth Trials Unit Collaboration and the main trial is managed by the Bristol Trials Centre Clinical Trials and Evaluation Unit and sponsored by Royal Brompton & Harefield NHS Foundation Trust. Each participant has the right to withdraw at any time. In addition, the investigator may withdraw the participant from their allocated treatment group if a clinical reason for not performing the surgical intervention is discovered. If a participant wishes to withdraw, any data already collected will be included in the

study analyses, unless the participant expresses a wish for their data to be excluded. Withdrawing patients will be asked if they would continue in follow-up and complete the requisite questionnaires. Participants who choose to withdraw from the study will be treated according to their hospitals' standard procedures.

The findings will be disseminated by usual academic channels, that is, presentation at international meetings and peer-reviewed publications. A full report for the funder will be written on completion of the study and a lay summary of the results provided to patients.

## Major changes to protocol

Since the first study protocol was approved by the Research and Ethics Committee (the current version is V.6.0, 10 April 2019), the following changes have been made:

▶ Qualitative assessment substudy added, as part of the pilot phase only.
▶ The EuroQol EQ-5D-5L was added.
▶ Updates to transition from pilot phase to main study, including addition of the integrated QRI and economic evaluation, and removal of the collection of blood and tissue samples, and one of the disease-specific questionnaires—the EORTC QLQ LC-13.
▶ Length of follow-up extended from 2 years until the end of the study for all participants to ensure that the study has 80% power.
▶ Video-recording aspect of the surgical quality assurance removed as this was deemed impractical by sites, and it was agreed that it was unnecessary by the Data Safety and Monitoring Committee and the Trial Steering Committee, acknowledging the other surgical quality assurance measures that are in place.

## Study progress

Recruitment started in May 2015 and 308 patients have been randomised so far (correct on 25 May 2020). A total of 66 patients from the pilot study are included in this figure. Recruitment will continue until 30 September 2020.

The full protocol is available online (https://www.journalslibrary.nihr.ac.uk/programmes/hta/1518831/).

**Author affiliations**
[1]Academic Division of Thoracic Surgery, Royal Brompton and Harefield NHS Trust, London, UK
[2]The Glenfield Hospital, University Hospitals of Leicester NHS Trust, Leicester, UK
[3]Mesothelioma UK, Glenfield Hospital, Leicester, UK
[4]Cardiothoracic Centre, Sheffield Teaching Hospitals NHS Foundation Trust, Sheffield, UK
[5]Population Health Sciences, University of Bristol, Bristol, UK
[6]Cancer Research UK Centre Leicester, University Hospitals of Leicester NHS Trust, Leicester, UK
[7]Department of Medicine, Royal Marsden NHS Foundation Trust, London, UK
[8]Department of Thoracic Oncology, Papworth Hospital NHS Foundation Trust, Cambridge, UK
[9]Department of Thoracic Surgery, Barts Health NHS Trust, London, UK
[10]The Beatson West of Scotland Cancer Centre, Gartnavel General Hospital, Glasgow, UK
[11]Thoracic Surgery, Guy's and Saint Thomas' Hospitals NHS Trust, London, UK

[12]Respiratory Medicine, South Tyneside NHS Foundation Trust, South Shields, UK

[13]Lung Cancer, Aneurin Bevan University Health Board, Newport, UK

[14]Oncology, Derby Teaching Hospitals NHS Foundation Trust, Derby, UK

[15]Department of Thoracic Surgery, Golden Jubilee National Hospital, Clydebank, UK

[16]Department of Oncology, New Cross Hospital, Wolverhampton, UK

[17]Oncology, South Tees Hospitals NHS Foundation Trust, Middlesbrough, UK

[18]Academic Respiratory Unit, School of Clinical Sciences, University of Bristol, Bristol, UK

[19]Respiratory Research Unit, North Bristol NHS Trust, Westbury on Trym, UK

[20]Thoracic Surgery Department, Leeds Teaching Hospitals NHS Trust, Leeds, UK

[21]Oncology, Colchester Hospital University NHS Foundation Trust, Colchester, UK

[22]Clatterbridge Cancer Centre, Clatterbridge Cancer Centre NHS Foundation Trust, Bebington, UK

[23]Plymouth Oncology Centre, University Hospitals Plymouth NHS Trust, Plymouth, UK

[24]Kent Oncology Centre, Maidstone and Tunbridge Wells NHS Trust, Maidstone, UK

[25]Department of Medical Oncology, Barts Health NHS Trust, London, UK

[26]The Oncology Care Team, Norfolk and Norwich University Hospital NHS Trust, Norwich, UK

[27]Medical Oncology, The Christie NHS Foundation Trust, Manchester, UK

[28]Heamatology and Oncology Unit, North West Anglia NHS Foundation Trust, Peterborough, UK

[29]Bristol Trials Centre (CTEU), University of Bristol, Bristol, UK

[30]Health Economics Research Centre, Nuffield Department of Population Health, University of Oxford, Oxford, UK

**Acknowledgements** The MARS 2 trial is sponsored by The Royal Brompton and Harefield NHS Foundation Trust. The sponsor will be responsible for the oversight of the MARS 2 study and to ensure that the trial is managed appropriately. We want to thank the large teams at each hospital who work on the MARS 2 study (representatives from each are listed below). We would also like to thank Dr Fiona McDonald from the Royal Marsden Hospital, Dr Nagmi Qureshi from Papworth Hospital and Professor Simon Padley from the Royal Brompton Hospital for their radiotherapy advisory roles. Thank you also to Professor Andrew Nicholson for his histopathology advisory role for MARS 2.

**Collaborators** MARS 2 Trialists: Project management team members: Athanasia Gravani, Holly McKeon, Wendy Underwood, Rachel Brophy, Nicola Farrar; Papworth Trials Unit Collaboration (pilot study): Victoria Hughes, Jane Elliott, Claire Matthews, Philip Noyes; Participating Sites Members: pilot study and main trial, University Hospitals of Leicester NHS Trust—medical and surgical site (opened 22/04/2015): Apostolos Nakas, Louise Nelson, Sheffield Teaching Hospitals NHS Foundation Trust—medical and surgical site (opened 13/05/2015): Sara Tenconi, Laura Socci, Hilary Wood, Helena Hanratty, Helena Stanley; South Tyneside and Sunderland NHS Foundation Trust—medical site (opened 19/06/2015): Judith Moore; Papworth Hospital NHS Foundation Trust—medical site (opened 01/07/2015): Robert Rintoul, Suzanne Miller, Amy Gladwell, Jenny Castedo, Amanda Stone; Colchester Hospital University NHS Foundation Trust—medical site (opened 03/11/2015): Charlotte Ingle, Hayley Hewer; South Tees Hospitals NHS Foundation Trust—medical site (opened 16/11/2015): Louise Li, Eleanor Aynsley, Andrea Watson, Charlotte Jacobs; The Clatterbridge Cancer Centre NHS Foundation Trust—medical site (opened 25/11/2015): Alison Hassall, Masuma Begum; University Hospitals of Derby and Burton NHS Foundation Trust—medical site (opened 26/11/2015): Christopher Worth, Ellie Piggott, Elizabeth Nadin; Leeds Teaching Hospitals NHS Trust—medical site (opened 02/12/2015): Victoria Ashford-Turner, Matthew Callister, Manchester University NHS Foundation Trust—medical site (opened 21/12/2015): Yvonne Summers, Raffaele Califano, Laura Cove-Smith, Matt Evison, Maria Blinston, Sara Waplington, Amal Ismail, Rachel Chant, Asmita Desai, Juliette Novasio, Marie Kirwan; The Royal Wolverhampton NHS Trust—medical site (opened 04/01/2016): Ian Morgan, Victoria Lake, Nichola Harris; Royal Gwent Hospital, Aneurin Bevan University Health Board—medical site (opened 08/02/2016): Simon Hodge; The Royal Marsden NHS Foundation Trust—medical site (opened 08/04/2016), Chelsea sub-site: Nadia Yousaf, Nadza Tokaca, Adam Januszewski, Avani Athauda, Anisha Ramessur, Emily Grist, Niamh Colman, Michael Flynn, Joan Joyce, Sarah Vaughan, Maria Piga, Derya Sahin, Agnieszka Yongue, Emma Turay, Sutton sub-site: Mary O'Brien, Jaishree Bhosle, Rajiv Kumar, Charlotte Milner-Watts, Jessica Brown, David Walder, Alexandros Georgiou, Xiaorong Wu, Naila Kaudeer, Kroopa Joshi, Michael Davidson, Shelize Khakoo, Bee Ayite, Kathryn Priest, James Dobbyn, Vasanthi Prathapan, Deborah McCrimmon, Natalie Ash, Alison Norton, Bianca Peet, Libby Hennessy, Rosemary Johnson, Laura White; Kingston sub-site: Edward Armstrong, Maria Coakley, Scott Shepherd, Narda Chaabouni, Katherina Sreter, Vasileios Angelis, Mariko Morishita, Jose Roca, Mary Jane de los Reyes Lauigan,

Katrin Sainudeen, Helen Morgan; Peterborough City Hospital, North West Anglia NHS Foundation Trust—medical site (opened 16/05/2016): Abigail Hollingdale, Chloe Eddings, Holly Warman; Participating Sites Members: main trial only, Barts Health NHS Trust—medical and surgical site (opened 05/06/2017): Jeremy Steele, Jo Hargrave, Resmi Jayachandran, Pratistha Panday, The Beatson West of Scotland Cancer Centre; Greater Glasgow Health Board—medical site (opened 14/07/2017): Austin McInnes; Golden Jubilee National Hospital—surgical site (opened 14/07/2017): Rocco Bilancia, Julie Buckley, Elizabeth Boyd; North Bristol NHS Trust—medical site (opened 28/02/2018): Natalie Zahan-Evans, Anna Morley; Norfolk and Norwich University Hospitals NHS Foundation Trust—medical site (opened 12/06/2018): Adela Dann, Eleanor Mishra, Pinelopi Gkogkou; University Hospitals Plymouth NHS Trust—medical site (opened 16/07/2018): Irene Harvey, Hilary Congdon; Barking, Havering and Redbridge University Hospitals NHS Trust—medical site (opened 24/07/2018): Alison Ray; Guy's and St. Thomas' NHS Foundation Trust—medical and surgical site (opened 10/08/2018): Jehan Mansi, Amy Quinn; Oxford University Hospitals NHS Foundation Trust—medical site (opened 19/10/2018): Najib M Rahman, Jack Seymour, Hannah Ball, Meenali Chitnis; Maidstone and Tunbridge Wells NHS Trust—medical site (opened 19/10/2018): Eirini Petroyannou, Kimberley Snoad, Monica Tavares Barbosa; University Hospitals Birmingham NHS Foundation Trust—medical site (opened 03/01/2019): Gary Middleton, Philip Earwaker, Haider Abbas, Parminder Sohal; Independent Trial Steering Committee members: Marcus Flather, Paul Beckett and Carol Tan have declared the following competing interest: Ethicon endostaplers—consultancies; Fergus Gleeson, Fergus MacBeth, Mavis Nye, Harvey Pass and Pauline Leonard have declared the following competing interests: Teva Amgen, Tom Treasure. Unless otherwise stated above, committee members have declared no competing interests, Independent Data Monitoring and Safety Committee members: Linda Sharples, Valerie Rusch, Mark Britton, Robin Rudd, Joseph S Friedberg, Peter Goldstraw. Unless otherwise stated above, committee members have declared no competing interests.

**Contributors** EL: Study design, preparation and drafting of protocol and manuscript, Chief Investigator for the trial. LD: Study design, preparation of protocol and review of manuscript. JE: Study design, preparation of protocol and review of manuscript, Principal Investigator and acquired data for the study. DE: Design of integrated qualitative study, preparation of study protocol, review of manuscript. DAF: Study design, preparation of protocol and review of manuscript, Principal Investigator and acquired data for the study. SP: Study design, preparation of protocol and review of manuscript, Principal Investigator and acquired data for the study. RCR: Study design, preparation of protocol and review of manuscript, Principal Investigator and acquired data for the study. DW: Study design, preparation of protocol and review of manuscript, Principal Investigator and acquired data for the study. CA, AB, LF, AI, MK, AK, PK, KL, TM, NAM, RM, DM TP, AR, RS, JS, ZT, PT, ST: Review of manuscript, Principal Investigator and acquired data for the study. KA: Study design, preparation and drafting of protocol and manuscript, oversaw study conduct and acquisition of data. RH: Statistical analysis plan, review of manuscript. KJ: Preparation and drafting of manuscript, oversaw study conduct and acquisition of data. BW: Drafting of manuscript, oversaw study conduct and acquisition of data. NM: Conduct of integrated qualitative study, preparation of study protocol, review of manuscript. EAS: Design of health economic component, preparation of study protocol, review of manuscript. CR: Study design, sample size and statistical analysis plan, drafting of protocol, review of manuscript.

**Funding** This research is funded by the National Institute for Health Research (NIHR) Health Technology Assessment (HTA) Programme (project number 15/188/31). The pilot study was funded by Cancer Research UK and Mesothelioma UK has contributed towards patient travel expenses. This study was designed and delivered in collaboration with the Bristol Trials Centre Clinical Trials and Evaluation Unit (CTEU), a UK Clinical Research Collaboration (UKCRC) registered clinical trials unit which, as part of the Bristol Trials Centre (BTC), is in receipt of NIHR Clinical Trials Unit (CTU) support funding. DE was supported by the NIHR Biomedical Research Centre at University Hospitals Bristol NHS Foundation Trust and the University of Bristol. SP acknowledges NHS funding to the Royal Marsden Hospital/ Institute of Cancer Research NIHR Biomedical Research Centre.

**Disclaimer** The views and opinions expressed therein are those of the authors and do not necessarily reflect those of the HTA programme, NIHR, NHS or the Department of Health and Social Care. The funder and sponsor approve any amendments to the study but have no direct involvement in study design; collection, management, analysis and interpretation of data; writing of the report; and the decision to submit this report for publication.

**Competing interests** None declared.

**Patient and public involvement** Patients and/or the public were involved in the design, conduct, reporting and dissemination plans of this research. Refer to the Methods section for further details.

**Patient consent for publication** Not required.

**Provenance and peer review** Not commissioned; externally peer reviewed.

**Open access** This is an open access article distributed in accordance with the Creative Commons Attribution 4.0 Unported (CC BY 4.0) license, which permits others to copy, redistribute, remix, transform and build upon this work for any purpose, provided the original work is properly cited, a link to the licence is given, and indication of whether changes were made. See: https://creativecommons.org/licenses/by/4.0/.

**ORCID iDs**
Barbara Warnes http://orcid.org/0000-0002-1326-0448
Elizabeth A Stokes http://orcid.org/0000-0002-4179-1369

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
