## [Reviewer comments · BMJ Open]

ARTICLE DETAILS

TITLE (PROVISIONAL)	Mesothelioma and Radical Surgery 2 (MARS 2): protocol for a multicentre randomised trial comparing (extended) pleurectomy decortication versus no (extended) pleurectomy decortication for patients with malignant pleural mesothelioma
AUTHORS	Lim, Eric; Darlison, Liz; Edwards, John; Elliott, Daisy; Fennell, DA; Popat, Sanjay; Rintoul, Robert; Waller, David; Ali, Clinton; Bille, Andrea; Fuller, Liz; Ionescu, Andreea; Keni, Manjusha; Kirk, Alan; Koh, Pek; Lau, Kelvin; Mansy, Talal; Maskell, Nick; Milton, Richard; Muthukumar, Dakshinamoorthy; Pope, Tony; Roy, Amy; Shah, Riyaz; Shamash, Jonathan; Tasigiannopoulos, Zacharias; Taylor, Paul; Treece, Sarah; Ashton, Kate; Harris, Rosie; Joyce, Katherine; Warnes, Barbara; Mills, Nicola; Stokes, Elizabeth; Rogers, Chris

VERSION 1 – REVIEW

REVIEWER	Andrea Wolf The Icahn School of Medicine at Mount Sinai, US
REVIEW RETURNED	28-Apr-2020

GENERAL COMMENTS	Outstanding description of an important study. Looking forward to this publication as well as the manuscript once trial is complete and results are analyzed. A few minor questions below: Page 8, lines 55-58- agree this would be the ideal standard definition as based on the reference cited but there is unfortunate continued lack of unanimity in nomenclature (although attempts such as Joseph Friedberg and IASLC committee's effort are being made to improve that for registries). It is probably therefore more accurate to say "Generally, pleurectomy/decortication..." Minor editorial revision- Page 9, line 5, would remove the "i)" or add a "ii)" to the second procedure listed. It should be noted that radical resection can be intended to be cancer-directed or have "curative" intent but may leave disease behind. Resection may involve removal of nearly all parietal/visceral pleura, with or without diaphragm and/or pericardium (or even EPP) and still leave disease behind. This is something in between the extended pleurectomy decortication described and the diagnostic or palliative type. Please clarify in the methods how survival will be calculated for each cohort (from date of diagnosis, date of start of chemotherapy, or date of surgery in surgical cohort).
---

	Please clarify that the analysis will be intention-to-treat (how will cross-over patients – can only occur for chemo patients who move to chemo + surgery—be counted?)
--	--

REVIEWER	Melissa Culligan University of Maryland School of Medicine Division of Thoracic Surgery Baltimore, Maryland United States of America
REVIEW RETURNED	05-May-2020

GENERAL COMMENTS	Congratulations to all members of the MARS 2 research team. This manuscript is well written and comprehensively outlines the conduct of the clinical trial. I have a few questions/comments to share with the authors. (1) Please be consistent throughout the manuscript when referring to the operation performed in this clinical trial. Rather than using "(extended) pleurectomy decortication" or "pleurectomy decortication", I would suggest "extended pleurectomy decortication" if this is in keeping with the intent of the operation. It is not clear as to why extended is in parentheses. (2) The summary of the major changes to the protocol section is excellent and will prove to be very valuable to those reading the results of the study when they are ready for publication. I would be in favor of building upon this section in the future as we all can learn from these protocol changes and apply those "lessons learned" to future protocol development. (3) Please add references to the abstract (4) The last sentence in the abstract should be revised as it is not accurate. There are other "radical treatments" currently under investigation for MPM, as an example the SMART Trial in Canada. (5) Eligibility question - Are patients who have received previous treatment for MPM eligible for enrollment?
---

VERSION 1 – AUTHOR RESPONSE

Response to the comments from Reviewer 1:

Q: Page 8, lines 55-58- agree this would be the ideal standard definition as based on the reference cited but there is unfortunate continued lack of unanimity in nomenclature (although attempts such as Joseph Friedberg and IASLC committee's effort are being made to improve that for registries). It is probably therefore more accurate to say "Generally, pleurectomy/decortication..."

A: Thank you, but we have stipulated this for the conduct of the trial and to inform the trial surgeons, and as we are nearly completed, we think that it would be important to preserve the statement as it stood at the outset of the trial. Also, we believe that we are correct to differentiate between pleurectomy/decortication and extended pleurectomy/decortication since these are the current internationally recognized definitions by the IASLC and IMIG.

Q. Minor editorial revision- Page 9, line 5, would remove the "i)" or add a "ii)" to the second procedure listed.

A. The i) has been removed from the Introduction, as requested.

Q. It should be noted that radical resection can be intended to be cancer-directed or have "curative" intent but may leave disease behind. Resection may involve removal of nearly all parietal/visceral pleura, with or without diaphragm and/or pericardium (or even EPP) and still leave disease behind. This is something in between the extended pleurectomy decortication described and the diagnostic or palliative type.

A. Thank you, very good point. However, for purpose of standardisation within the trial, we sought to define the objective of the surgery which is to remove all macroscopic evidence of disease.

Q. Please clarify in the methods how survival will be calculated for each cohort (from date of diagnosis, date of start of chemotherapy, or date of surgery in surgical cohort).

A. Thank you, survival is calculated from the time of randomisation (which occurs after two cycles of chemotherapy). This is now clarified in the Methods section of our revised manuscript.

Q. Please clarify that the analysis will be intention-to-treat (how will cross-over patients – can only occur for chemo patients who move to chemo + surgery—be counted?)

A. Thank you, as stated in our Statistical Analysis section, the analyses will be calculated by intention to treat and each person will be analysed in the arm to which she or he was allocated. It is interesting to note, that within MARS 2, cross overs were bilateral, a number of patients randomised to surgery did not wish to proceed with the operation.

Response to Comments from Reviewer 2:

(1) Please be consistent throughout the manuscript when referring to the operation performed in this clinical trial. Rather than using "(extended) pleurectomy decortication" or "pleurectomy decortication", I would suggest "extended pleurectomy decortication" if this is in keeping with the intent of the operation. It is not clear as to why extended is in parentheses.

A. Thank you, good point, we refer to the procedure used in this trial as (extended) pleurectomy decortication, because the aim of surgery was to remove all macroscopic disease and this can be achieved with either pleurectomy decortication (where the pericardium and diaphragm are left intact) or extended pleurectomy decortication (where either the pericardium or diaphragm are resected and reconstructed). The extended is in parentheses to indicate that either operation type could be used in MARS 2 to remove all macroscopic disease, depending on the individual case. We have amended the manuscript to make this clearer.

(2) The summary of the major changes to the protocol section is excellent and will prove to be very valuable to those reading the results of the study when they are ready for publication. I would be in favor of building upon this section in the future as we all can learn from these protocol changes and apply those "lessons learned" to future protocol development.

A. Thank you for your kind comments

(3) Please add references to the abstract

A. Thank you, we have been informed by the editorial team that this is not required.

(4) The last sentence in the abstract should be revised as it is not accurate. There are other "radical

treatments" currently under investigation for MPM, as an example the SMART Trial in Canada.

A. Thank you, but we are referring to the setting of a randomized clinical trial, and at present, patients participating in the SMART Trial in Canada are not randomized.

(5) Eligibility question - Are patients who have received previous treatment for MPM eligible for enrollment?

A. Thank you for your question. Yes, patients who have received previous treatment for MPM are eligible for enrolment in this study.

VERSION 2 – REVIEW

REVIEWER	Andrea Wolf The Icahn School of Medicine at Mount Sinai New York, NY United States
REVIEW RETURNED	22-Jun-2020
GENERAL COMMENTS	Well done!